# Next-Generation Sequencing to Elucidate the Semen Microbiome in Male Reproductive Disorders

**DOI:** 10.3390/medicina60010025

**Published:** 2023-12-22

**Authors:** Rhianna Davies, Suks Minhas, Channa N. Jayasena

**Affiliations:** 1Department of Metabolism, Digestion and Reproduction, Imperial College London, London W12 0HS, UK; rhianna.davies09@imperial.ac.uk; 2Department of Urology, Charing Cross Hospital, Imperial College NHS Trust, London W6 8RF, UK; suks.minhas@nhs.net

**Keywords:** semen, microbiota, reactive oxygen species, DNA damage, morphology

## Abstract

Mean sperm counts are declining at an accelerated rate and infertility is increasingly becoming a public health concern. It is now understood that human semen, previously considered to be sterile, harbours its own specific microbiome. Via activated leucocytes and the generation of reactive oxygen species, bacteria have the capability of evoking an immune response which may lead to sperm damage. Men with infertility have higher rates of both reactive oxygen species and sperm DNA damage. Due to the lack of sensitivity of routine culture and PCR-based methods, next-generation sequencing technology is being employed to characterise the seminal microbiome. There is a mounting body of studies that share a number of similarities but also a great range of conflicting findings. A lack of stringent decontamination procedures, small sample sizes and heterogeneity in other aspects of methodology makes it difficult to draw firm conclusions from these studies. However, various themes have emerged and evidence of highly conserved clusters of common bacteria can be seen. Depletion or over-representation of specific bacteria may be associated with aberrations in traditional and functional seminal parameters. Currently, the evidence is too limited to inform clinical practice and larger studies are needed.

## 1. Introduction

Infertility affects 10–15% of couples; about half of all cases are caused by male factors [1]. Whilst many factors are known to affect male fertility, including genetic abnormalities, environmental exposures, lifestyle, medical conditions, medication and trauma, 30–70% of cases of male infertility are unexplained [2,3]. Unexplained or idiopathic male-factor infertility, therefore, has no targeted therapy. The human body contains more bacteria than human cells [4]. Increasingly, the role of the human microbiome in reproductive health and disease is being investigated [5]. Dysbiosis of the female reproductive tract has been associated with adverse pregnancy outcomes and impaired response to fertility treatment [6,7]. Some authors have described an association between the presence of certain micro-organisms and specific aberrations in seminal parameters [8,9,10,11]. Though, as in other areas of fertility research, the male reproductive tract microbiota has been relatively neglected [12]. In the absence of pharmacotherapies to improve semen quality, understanding the seminal microbiome has become a priority. 

## 2. The Human Microbiome and Dysbiosis

The human microbiome describes a complex ecosystem of co-evolved organisms including bacteria, viruses, fungi and protozoa [13]. This ecosystem is understood to play a crucial role in many physiological processes in many body systems [14]. For example, gut bacteria are involved in digestion by the release of enzymes capable of breaking down complex carbohydrates to enhance absorption, fermentation and biosynthesis of essential vitamins [15,16]. The gastrointestinal tract contributes 29% of the human microbiome; the genito-urinary tract contributes 9% [17]. The gastrointestinal, dermatological, respiratory and female genito-urinary-system microbiomes have been particularly investigated [18,19,20,21]. Dysbiosis in these systems has been linked with irritable bowel syndrome, allergy, chronic obstructive pulmonary disease, and impaired fertility and preterm birth respectively [18,19,20,21,22,23]. In the female, *Lactobacillus* has been identified as pivotal to maintain equilibrium of the resident vagina flora [24]. When *Lactobacillus* is depleted in the endometrium, implantation failure is more likely [6]. 

## 3. Male Fertility

Human spermatozoa, released from the testes, are bathed in secretions as they travel down the epididymis [25]. These secretions arise from the accessory glands and include the seminal vesicles, bulbourethral glands and prostate [26]. Seminal fluid, made up of sperm and secretions, has an alkaline pH [25]. It is understood that proteins from the seminal fluid are actively imported into the sperm [27]. The vagina has an acid pH and, following ejaculation, exposure to this environment induces capacitation. Capacitation describes the enzymatic activation that allows the sperm to enter the female oocyte [25]. Upon binding, the sperm undergoes the acrosome reaction, with further enzymatic activation facilitating burrowing through the zona pellucida and binding to the oocyte plasma membrane [25]. It was previously understood that human sperm contributed only its DNA to the embryo; however, it is now known to play a crucial role in epigenetic modifications and placental formation [26]. Thus, fertilisation and the associated post-fertilisation events require a complex stepwise process dependent upon functional sperm [25]. Spermatic aberrations can lead to unequal embryonic cleavage, failed development of the blastocyst, inadequate implantation and miscarriage [28]. 

There has been an overall decrease in mean sperm count of 62.3% since 2000 [29]. This equates to a 2.64% decline per annum and represents a significant global health concern [29]. With male factors accounting for half of all cases of infertility, there is an urgent need for evidence-based therapies to improve sperm quality. Semen analysis remains the cornerstone of assessment of male reproductive capacity; however, it does not provide information regarding function [30]. Damage to the DNA carried by the sperm and levels of reactive oxygen species, known to induce DNA damage, can now be tested [31,32,33]. Whilst these newer methods of assessing the sperm are commercially available, there are no current recommendations to guide clinical use [1]. Symptomatic infection is an established cause of male infertility [1,34]. The inflammatory response to infection may disrupt the environment of the reproductive tract at various stages, including those that take place in the epididymis, accessory glands and testes [35,36]. Furthermore, the sperm themselves can be damaged at various stages of development [35]. Common symptomatic micro-organisms affecting male reproduction include *Chlamydia trachomatis* and *Neisseria gonorrhoeae* [1,34,35,37]. Additionally, non-sexually-transmitted infections such as *Escherichia coli* may deleteriously impact fertility potential. Infections of the male reproductive tract are understood to be responsible for about 15% of male factor infertility [37]. However, the current European Association of Urology guidance states that, whilst antibiotics may improve the overall quality of the spermatozoa, there is no evidence of increased pregnancy rates after antibiotic treatment of the male partner [1,34]. 

Furthermore, asymptomatic bacteriospermia has been reported to exist in 33% of infertile men; however, other studies report rates as high as 70% [38,39]. Within the semen exists an immune system primed to react to infection [36]. Seminal leukocytes generate reactive oxygen species (ROS) in response to infection [40]. Paradoxically, ROS, alongside bactericidal actions, also damage sperm DNA; and, thus, the functional capacity of the sperm [40]. Additionally, damaged sperm release ROS, leading to a cycle of oxidative stress-induced damage [40]. A study by Moretti et al. identified that 15% of men presenting with subfertility for semen analysis, despite being asymptomatic of genito-urinary infection, had leukocytospermia [41]. It is presumed that leukocytospermia may be a surrogate marker for bacteriospermia [42]. Multiple studies show that men in couples affected by infertility or recurrent pregnancy loss have higher rates of elevated seminal ROS and sperm DNA fragmentation [33,43,44,45,46]. However, Sanocka-Maciejewska et al. reported that the bacteria most commonly isolated from the general fertile population have no impact on sperm quality [47]. 

## 4. The Human Seminal Microbiome

Seminal fluid, with an alkaline pH; enriched with lipids, sugars and proteins to nourish and protect the sperm; provides an optimal environment for micro-organisms [48,49]. The male genital tract has previously been considered sterile; indeed, the presence of any bacteria was considered pathological [50,51]. The culture or PCR-based methods used to inform this opinion are fundamentally flawed due to their inability to culture all bacteria [52]. More recently, advanced next-generation sequencing techniques (NGS) have allowed analysis of the seminal microbiome and identification of its unique composition [8,9,10,11] (Table 1). The exact origins of the bacteria within human seminal fluid remain unclear, and it is not known whether they represent transient colonisation or a static resident flora. Whilst similarities between the seminal and urinary microbiome exist, which is unsurprising given their common urethral tract, both the total count and range of bacteria is higher in semen compared with that in urine [51]. Furthermore, only 1/3 of the micro-organisms are shared, suggesting that the upper genital tract significantly contributes to the seminal microbiome [51]. There is likely to be reciprocal transfer of the microbiome between sexual partners. A study by Mandar et al. comparing the vaginal microbiome before and after sexual intercourse established that the seminal microbiome has a considerable effect on the constitution of the vaginal microbiome [53]. *Lactobacillus crispatus* abundance decreased post-coitally in response to the seminal microbiome [53]. Furthermore, the female sexual partners of men with leukocytospermia were more likely to be dominated by *Gardnerella vaginalis* [53].

## 5. Next-Generation Sequencing of the Human Seminal Microbiome

NGS describes the simultaneous assessment of millions of genetic fragments. It is capable of sequencing thousands of genes in a short period of time [59]. There are four main steps to the sequencing protocol: fragmentation, library preparation, sequencing and analysis. The DNA or RNA is fragmented into segments using mechanical, sonic or enzymatic methods. These segments are organized into a library for analysis. Bioinformatic programmes then compare these ‘reads’ to the reference genome (Figure 1). For NGS of the microbiome 16S RNA with 6 hypervariable regions, (V1–6) is used as the 16s RNA gene present in all bacteria with regional sequence variation to characterise species [8,9,10,11].

### 5.1. In Male Infertility

Molina et al. assessed the V3–4 hypervariable region of 16S RNA via NGS in testicular samples obtained by open-testicular biopsy and found the testes not to be sterile [60]. They recruited men from an assisted reproduction clinic in Spain who presented with azoospermia (no sperm in the ejaculate), oligoasthenoteratozoospermia (sperm that is abnormal in low in count, or poor motility and abnormal morphology) or DNA fragmentation. They collected a total of 307 testicular spermatozoa from 24 semen samples collected from a total of 11 men. To avoid contamination, the scrotum was cleaned using an antiseptic solution prior to biopsy in an air-purified theatre. Aside from preparation of the skin prior to incision to avoid contaminating the sample prior to collection, post-collection decontamination procedures were also followed. They performed both Decontam and MicroDecon. Decontam describes the identification of contaminant reads based on their occurrence in the case sample vs. a control sample; a prevalence threshold is then set. MicroDecon is based on the principle that both a case and a control sample will be equally exposed to environmental contaminants [61]. Thus, the proportion of contaminants found in the control sample will also be removed from the case sample [62]. Following decontamination, 10 specific bacteria were identified as specific to testicular sperm. This included Clostridium and Prevotella [60]. The notion that the testes are not sterile is supported by Alfano et al., who performed a cross-sectional study that investigated the testicular microbiome in azoospermic men following microdissection testicular sperm extraction (microTESE) [63]. Men with idiopathic non-obstructive azoospermia (NOA) had a greater bacterial load compared with normospermic men, with a predominance of *Actinobacteria* and *Firmicutes*. The samples from men with idiopathic NOA had a decreased taxa richness secondary to depletion of *Bacteroidetes* and *Proteobacteria* [63]. Their methodology describes sample collection and handling under sterile conditions, but no decontamination protocol [63]. Campbell et al. undertook NGS analysis of seminal microbiome taxonomy in men with NOA (n = 14) compared with fertile controls (n = 19). They found that alpha diversity was similar between the groups. At genera level, *Escherichia, Shigella, Sneathia and Raoutella* differed significantly between groups [64]. Chen et al. also investigated the microbiomes of men with NOA (n = 30) and normal controls (n = 30) via NGS of the V3-V4 16S RNA hypervariable region [9]. The hands and penis were washed thoroughly with warm soapy water and 75% alcohol for disinfection prior to collection. They set up three negative controls to exclude potential contaminants from air, tabletop and reagents. Contaminants in 16s RNA sequences were removed using the Decontam R package [9]. In agreement with Campbell et al., alpha diversity was similar between the groups; however, they found significant differences between groups at genera level; *Ruegeria* and *Donghicola* dominated the NOA group. Arguably, in the case of NOA, subfertility is possibly accounted for by pathology other than the seminal microbiome and, as such, these studies should be interpreted with caution. 

In Switzerland, Baud et al. evaluated the bacterial composition of seminal fluid and its impact on sperm parameters in 26 men with normal sperm and 68 men with at least one abnormality in semen analysis [11]. They assessed the V1–V2 hypervariable region of 16S RNA. During their extraction and library preparation, they also processed two samples of sterile water to function as extraction negative controls. They reported semen samples broadly clustered into three microbiota profiles: (1) *Prevotella*-enriched, (2) *Lactobacillus*-enriched and (3) *Polymicrobial*. In Baud et al., *Prevotella*-enriched samples had the highest bacterial load (*p* < 0.05) but there was no difference in microbial richness or alpha diversity between clusters. The authors concluded that there was no correlation between seminal microbiome and seminal parameters, suggesting that the microbiome may not play a major role in male infertility [11]. However, differential abundance testing differential abundance testing found three specific genera that were significantly depleted or enriched in some of the sperm quality groups (*p* < 0.05); *Prevotella* was enriched in the samples defective sperm motility, and *Staphylococcus* and *Lactobacillus* were enriched in the group with normal sperm morphology. Thus, at genera level, differential abundance suggested that a small subset of microbes may impact motility and morphology [11]. Weng et al. identified 3 groups within 96 Taiwanese men within a fertility clinic: (1) *Pseudomonas*-predominant, (2) *Lactobacillus*-predominant and (3) Prevotella [8]. Of the controls, 80.5% were *Lactobacillus*-predominant [8]. Hou et al. examined the samples from 19 healthy sperm donors and 58 infertile men with seminal abnormalities, divided into those with asthenozoospermia, oligoasthenozoospermia and severe oligoasthenozoospermia and azoospermia, in China [54]. Their methodology included washing the hands and thoroughly cleaning the penis with 75% alcohol prior to sample production. They found a wide range of bacteria in the semen of infertile men and controls. In contrast to other studies, they identified six clusters, the most common (23.4%) characterised by high proportions of *Streptococcus*, *Corynebacterium, Finegoldia* and *Veillonella*. They identified no overall pattern between community composition in infertile men compared with that in controls; however, they did report a significant negative association between sperm quality and the presence of *Anaerococcus*. Further studies are required to establish if this is a causal link. In America, Lundy et al. compared the microbiological composition of the gut, semen and urine of men with primary idiopathic infertility (n = 25) and healthy men with proven paternity (n = 12) using the V3–4 hypervariable region of 16S RNA [58]. Semen samples were collected via masturbation, urine samples by midstream catch and rectal samples via swab using an aseptic technique. A series of positive and negative controls were set up to identify potential sample contaminants. The reads for negative controls were subtracted from the samples during bioinformatics analysis. Negative controls included those for sample collection (semen, urine, rectal swab), code extraction (e.g., reagent), preparation and sequencing. The ZymboBIOMICS (Zymo Researc, Irvine, CA, USA) mock community standard was used as a positive control [58]. This describes a well-characterised sample comprising gram-positive and gram-negative bacteria with various defined characteristics. Like in other studies, Lundy et al. identified a diverse seminal microbiome; though, in contrast to many studies, alpha diversity was greater in the infertile group. *Anaerococcus* was enriched in the semen of infertile men [58]. *Prevotella* abundance was inversely associated with sperm concentration, whilst *Pseudomonas* was directly associated with total motile sperm count. There were similarities between the semen and urinary microbiomes, likely due to the urethral contribution to the semen sample [58]. Vasectomy appeared to alter the seminal microbiome, suggesting a testicular or epididymal contribution, though the study was not powered to investigate this. It is noteworthy that elevated seminal ROS in the context of a varicocele is an established phenomenon [65,66]. Garcia-Segura studied the seminal microbiome in a Spanish population and evaluated its relationship to functional seminal parameters, i.e., DNA damage and oxidative stress [55]. They enrolled 14 healthy normospermic men and compared their semen samples to those of 42 infertile men with normospermia (idiopathic male infertility). There were methods in place for cleaning of the hands and penis prior to sample production. A sterile swab was moved around in the collection room to establish the composition of environmental contaminants. The entire hypervariable region (V1–9) of 16S RNA was analysed. The most abundant genera were *Finegoldia, Peptoniphilus, Anaerococcus, Campylobacter, Streptococcus, Staphylococcus, Moraxella, Prevotella, Ezakiella, Corynebacterium* and *Lactobacillus*. Of note, *Ezakiella* has not previously been described within the seminal microbiome. At genera level, samples enriched with *Moraxella, Brevundimonas* and *Flavobacterium* negatively correlated with the sperm global DNA fragmentation; *Brevundimonas* and *Flavobacterium* associated with higher sperm motility; *Brevundimonas* associated with lower oxidative-reduction potential; and *Actinomycetaceae, Ralstonia* and *Paenibacillus* correlated with reduced chromatin protamination status and increased sperm DNA fragmentation. Veneruso et al. compared the seminal microbiome of men with semen abnormalities compared with that of healthy controls (n = 20) presenting for fertility assessment in Italy [56]. They used the V3–4 hypervariable region of 16S RNA for NSG. To reduce contamination, extraction was performed under a laminar-flow hood and two negative controls were processed to identify environmental contributions. They found a reduced bacterial richness in infertile men compared with that in controls. The samples of the infertile group were enriched in certain genera including *Mannheimia, Escherichia, Shigella* and *Varibaculum.* The authors concluded that reduced bacterial richness in the infertile group suggests that poor semen quality is associated with reduced bacterial biodiversity and an unequal representation of the different taxa [56]. Yang et al. compared the seminal microbiome of men with oligoasthenospermia (n = 22), asthenospermia (n = 58), azoospermia (n = 8), oligospermia (n = 13) and healthy controls (n = 58) to establish if different abnormalities in seminal parameters have an associated colonisation patter [57].Their methodology included assessing the V1–2 hypervariable region of 16S RNA and a cleaning regime prior to sample collection. They found that the seminal microbiome of men with asthenospermia and oligoasthenospermia were significantly different compared with samples from healthy controls. They were enriched with *Ureaplasma, Bacteroides, Anaerococcus, Finegoldia, Lactobacillus* and *Acinetobacter lwoffii*. The authors concluded that the potential use of specific micro-organisms as biomarkers for these semen parameter abnormalities warrants further investigation [57]. 

### 5.2. In Assisted Reproductive Technique Outcomes (ART)

ART has revolutionised the management of infertility. However, the success rate per transferred embryo remains low and failures of implantation are often unexplained. Only a handful of studies have assessed the seminal microbiome in relation to outcomes during ART [67,68,69]. IVF largely bypasses the natural immune defence mechanisms of the female vaginal microbiome against the microbiome of the sperm [70]. As such, IVF protocols include microbial decontamination methods and often infuse their culture medium with antibiotics [71]. Okwelogu et al. used NGS of the V4 hypervariable region of 16S RNA to correlate the composition of the seminal and vaginal microbiome with IVF outcomes in 36 couples [67]. Cleaning and decontamination methods were not described in the methodology. Seminal fluid microbiota compositions had lower bacterial concentrations compared with those in the vagina but significantly higher species diversity. Both *Mycoplasma* and *Ureaplasma* were enriched in azoospermic men. In normospermic semen, *Lactobacillus* (43.86%) was the most abundant, followed by *Gardnerella* (25.45%); in the corresponding vaginal samples, *Lactobacillus* (61.74%) was the most abundant, followed by *Prevotella* (6.07%) and *Gardnerella* (5.86%). Semen samples with positive IVF outcomes were significantly colonized by *Lactobacillus jensenii* and *Faecalibacterium*, and significantly less colonized by *Proteobacteria, Prevotella* and *Bacteroides* [67]. The authors concluded that seminal samples with positive IVF have a different pattern of colonisation than those with lower success and raised the possibility of lactobacillus supplementation in association with IVF [67]. Štšepetova et al. studied 50 infertile couples to determine the prevalence and counts of bacteria in IVF samples via NGS of the V2–3 hypervariable region of 16S RNA [68]. Prior to sample collection, men were asked to wash their hands and penis, though no precautions were undertaken to remove or identify environmental contaminants [68]. It is noteworthy that the female partner had undergone a transvaginal scan prior to egg collection; it is not impossible that this altered the vaginal flora. The results concluded that IVF does not occur in a sterile environment. The presence of *Staphylococcus* and *Alphaproteobacteria* was associated with clinical outcomes such as sperm and embryo quality [68]. Koort et al. examined the seminal and vaginal microbiome via NGS of the V6 hypervariable region of 16S RNA in 97 couples undergoing ART and 12 healthy couples in Estonia [69]. The methodology described a cleaning regime prior to sample production but no sample decontamination. The men with *Acinetobacter*-associated community who had children in the past had the highest ART success rate. This was in contrast to their female partners, for whom it was found that *Lactobacillus iners*-predominant and *Lactobacillus gasseri*-predominant microbiome had a lower ART success rate than it did in women with the *Lactobacillus crispatus*-predominant or the mixed lactic-acid-bacteria-predominant type [69]. In summary, studies investigating ART success and microbiome show that IVF does not occur in a sterile environment, and it is likely that both the vaginal and seminal microbiome affect IVF success. 

### 5.3. Limitations

There is great heterogeneity between the findings of these studies, partly due to methodological differences. Many were limited by virtue of small study size and being single-institution studies. Such small samples allow little accountability for demographical or lifestyle contributions. It is known, for example, in research on the gut microbiota, that it is affected by age, ethnicity, diet, systemic conditions and genetics [15,23,58]. Sexual activity and circumcision, for example, are likely to affect the seminal microbiome; the microbiota of heterosexual and homosexual men has been shown to differ [53,72,73]. Furthermore, the seminal microbiome is considered a low biomass sample; as such, contaminants will have a relatively large impact, resulting in erroneous over-reporting of bacterial contribution. A number of studies performed stringent decontamination procedures and exclusionary environment sampling, but many didn’t. 

## 6. Conclusions

Infertility is increasingly becoming a public health concern. The burden of research, investigation and management has historically fallen on the female in the couple. More recently attention has turned to the male partner. A role for bacteria in the activation of leukocytes, induction of ROS, damage of sperm and, thus, further ROS activation has been postulated [34,35,36,37]. Culture and PCR methods are unable to identify all organisms present and, thus, do not offer accurate representation of the microbiome [52]. Next-generation RNA sequencing describes the simultaneous assessment of millions of genetic fragments; for sequencing of the microbiome, the highly conserved 16S RNA region is used [10,74,75,76,77,78]. A mounting number of studies have assessed the seminal microbiome in health and disease using NGS [10,74,75,76,77,78]. There is increasing evidence of consistent clusters of bacterial species within semen, and over- or under-representation of specific taxa may impair traditional and functional seminal parameters. There is a great range of conflicting associations amongst the literature, likely due to the heterogeneity of study design. Currently, the evidence is inadequate to warrant routine testing of the seminal microbiome in asymptomatic infertile men.

## 7. Future Directions

Greater knowledge of the impact of the seminal microbiome on semen quality may allow for targeted therapies. Additionally, it remains to be seen whether manipulation of the microbiome with either probiotics or antimicrobials would have an impact on fertility outcomes. Repeated temporal sampling of men is also required to determine whether bacteria have transient or persisting presence within the semen. Finally, the likelihood that partners will have interdependent microbiota means that joint investigation and/or treatments targeting the microbiome are likely to be required to achieve meaningful changes to fertility outcomes.

## Figures and Tables

**Figure 1 medicina-60-00025-f001:**
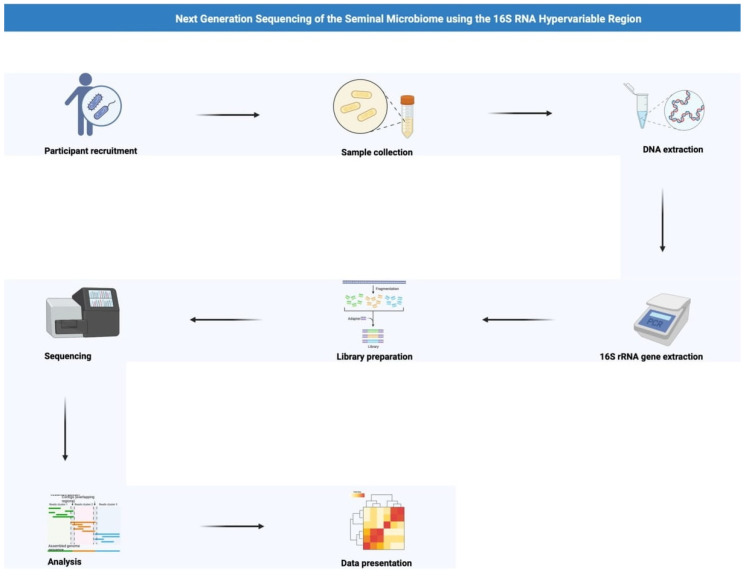
A schematic representation of next generation sequencing of the seminal microbiome using the 16S RNA hypervariable region. Created using BioRender.com.

**Table 1 medicina-60-00025-t001:** Summary of next-generate RNA sequencing of the human seminal microbiome.

Author	Country	N	Hypervariable Region *	Main Findings
Baud et al. [11]	Switzerland	94	V1–2	Normal sperm morphology: ↑*Lactobacillus* Abnormal sperm motility: ↑*Prevotella*
Weng et al. [8]	Taiwan	96	V1–2	Normal sperm parameters: ↑*Lactobacillus;* ↑*Gardnerella* Abnormal sperm parameters: ↑*Prevotella*
Hou et al. [54]	China	77	V1–2	Abnormal sperm parameters:↑*Anaerococcus*
Garcia-Segura et al. [55]	Spain	56	V1–9	Increased sperm global DNA fragmentation: ↑*Moraxella;* ↑*Brevundimonas;* ↑*Flavobacterium* Reduced chromatin protamination status and increased double-stranded DNA fragmentation: ↑*Actinomycetaceae;* ↑*Ralstonia;* ↑*Paenibacillus* Higher sperm motility: ↑*Brevundimonas;* ↑*Flavobacterium* Lower oxidative-reduction potential: ↑*Brevundimonas*
Veneruso et al. [56]	Italy	20	V3–4	Abnormal sperm parameters: ↑*Mannheimia;* ↑*Escherichia;* ↑*Shigella;* ↑*Varibaculum*
Yang et al. [57]	China	159	V1–2	Asthenospermia and oligoasthenospermia: ↑*Ureaplasma;* ↑*Bacteroides;* ↑*Anaerococcus;* ↑*Finegoldia;* ↑*Lactobacillus;* ↑*Acinetobacter lwoffii*
Lundy et al. [58]	USA	37	V3–4	Idiopathic infertility group: ↑*Aerococcus* ↑*Prevotella* was inversely associated with sperm concentration Total motile count was directly associated with ↑*Pseudomonas*

↑ = samples enriched; ↓ = samples deplete; * Hypervariable region of 16S RNA analysed via next-generation sequencing.

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
