# Peer review of "Next-Generation Sequencing to Elucidate the Semen Microbiome in Male Reproductive Disorders"

_medicina, 2023, doi:10.3390/medicina60010025_

Round 1

Reviewer 1 Report

Comments and Suggestions for Authors

Next generation sequencing to elucidate the semen microbiome 2 in male reproductive disorders.

In the present study, a review about the employ of next generation sequencing technology to characterise the seminal microbiome is carried out.

The article is scientifically very interesting; however, it has some minor aspects that need to be corrected.

Introduction. The introduction described the well state of the art of study problem.

Line 44: Chlamydia trachomatis and Neisseria gonorrhoeae

Line 183-185: Gardnerella vaginalis

Better that the name of the various microbial agents should be written in italics.

Line 80: 4. Next generation sequencing of the human seminal microbiome. Adding a subheading in different sections will make the article clearer and more organized.

Do you think it would be appealing to add any reference about the use of the Stammey-Meares test?

References. I have been unable to find the reference in the bibliography, and they don't follow the numbering of the bibliography.

 Line 210, references 52-62.

Line 212, references 63-64

Comments on the Quality of English Language

Minor editing of English language required.

Author Response

Thank you for your valuable comments on the manuscript. Please find attached word document addressing them

Reviewer 2 Report

Comments and Suggestions for Authors

The paper investigates on a very significant issue i.e., the influence of the semen microbiome in male fertility which is intrinsically novel, given the fact that the role of bacteria in male reproduction is a less explored area. The utilization of next-generation sequencing technology for characterizing seminal microbiome shows novelty in terms of leveraging technology for understanding health concerns. The paper provides a potentially important contribution in the area of reproductive health, with potential applications in clinical practices. The manuscript is well organized, with methodological coherence which lends credence to the findings presented. It bridges the intersection of health science and machine learning, which could spur exciting new endeavors or investments in the space.

Potential revise    - Lack of methodological rigor:        - More details should be provided on methods of decontamination and environment sampling.        - The methodology should more clearly detail the process of sequencing technology.    - Lack of data presentation and discussion:        - A described Table A or other figures on data should be added.        - The discussion and interpretation of the findings could be more thorough.    - Lack of an algorithm or model development:        - The manuscript does not discuss any specific ML model or algorithm employed for the next-generation sequencing.        - The results of the ML approach used could have been evaluated in comparison to other state-of-the-art approaches.    - Lack of validation of findings:        - The lack of repeated temporal sampling or multi-centred approach could raise questions about the validity of findings.

Suggestions for Improvement    - More detailed descriptions of methods, especially the next-generation sequencing technology, should be added to ensure replication and validity of result.    - The paper could benefit from the addition of figures and tables that show the actual data generated by the sequencing.    - Consider incorporating a specific ML model or approach for the next-generation sequencing to lend more relevance for the ML community.    - The manuscript could benefit from a more extensive validation setting, like temporal sampling or multi-centred studies.

Author Response

(The authors gave the same response as above.)
